# Antibiotic De-Escalation in the Intensive Care Unit: Rationale and Potential Strategies

**DOI:** 10.3390/antibiotics14050467

**Published:** 2025-05-03

**Authors:** Sarah Singer Matuszak, Lauren Kolodziej, Scott Micek, Marin Kollef

**Affiliations:** 1Department of Pharmacy, Barnes-Jewish Hospital, St. Louis, MO 63110, USA; sarah.matuszak@bjc.org (S.S.M.);; 2Department of Pharmacy Practice, University of Health Sciences and Pharmacy, St. Louis, MO 63110, USA; scott.micek@uhsp.edu; 3Division of Pulmonary and Critical Care Medicine, Washington University School of Medicine, St. Louis, MO 63110, USA

**Keywords:** antibiotics, sepsis, antimicrobial stewardship, duration, spectrum, de-escalation, intensive care unit, antimicrobial resistance, rapid diagnostics

## Abstract

Antibiotic de-escalation (ADE) is important to help optimize antibiotic use and balance the positive and negative effects of antimicrobial therapy. ADE should be performed promptly, and infections should be treated with the shortest course of antimicrobials as clinically feasible to avoid unnecessary use of broad-spectrum antimicrobials. Several tools have been developed to increase efficient ADE, including rapid diagnostic tests (ex. multiplex PCR), MRSA nasal PCR/culture, and biomarkers. Multiplex PCR and MRSA nasal PCR/culture have been associated with reductions in inappropriate antibiotic use. Procalcitonin, a biomarker, has been associated with shorter antimicrobial durations in some studies; however, widespread use may be limited by lack of specificity for bacterial infections, cost, and lack of set cut-off points. Additional biomarkers such as IL-6, HMGB1, presepsin, sTREM-1, CD64, PSP, proadrenomedullin, and pentraxin-3 are currently being studied. As technology improves, additional tools may be leveraged to better optimize ADE even better, such as antimicrobial spectrum scoring tools and artificial intelligence (AI). Spectrum scores, which quantify antibiotic activity using specific numeric values, could be incorporated into electronic health records to identify patients on unnecessarily broad antibiotics. AI modeling has the potential to predict personal antibiograms or provide the probability that an empiric regimen may cover a particular infection, among other potential applications. This review will discuss the literature associated with ADE in the ICU, selected tools to help guide ADE, and perspectives on how to implement ADE into clinical practice.

## 1. Introduction

Infection is common in patients admitted to intensive care units (ICUs) worldwide and is associated with increased mortality [1]. Moreover, infection with antibiotic-resistant organisms and/or ICU-acquired infections further increase this risk. In observational cohort studies of patients with septic shock, delays in administration of appropriate antibiotic therapy have consistently been associated with increased mortality [2,3,4,5,6]. Given these data, rapid prescribing of broad-spectrum antibiotics is recommended by international guidelines, and implementation has been widely adopted in patients with both sepsis and septic shock [7]. A cross-sectional study of nearly 900,000 adults with community-onset sepsis in 241 United States hospitals reported that 65.1% received empiric therapy against methicillin-resistant *Staphylococcus aureus* (MRSA) or an anti-pseudomonal beta-lactam [8]. However, 90.5% of patients treated with these agents did not have resistant Gram-positive or Gram-negative organisms isolated. Unfortunately, there are many possible adverse consequences of empiric broad-spectrum antibiotics, including *Clostridioides difficile* infection (CDI), acute kidney injury, hematologic or neurologic toxicity, disruption of the gut microbiome, and development of antimicrobial resistance (AMR) [7,9]. To balance the positive and negative effects of broad-spectrum therapy, antibiotic de-escalation (ADE) has been promoted for a quarter century. This paper reviews the literature associated with ADE from a critical care lens, including when to assess for opportunities for ADE, use of de-escalation tools (including rapid diagnostics and biomarkers), antimicrobial durations, and potential future directions for ADE.

## 2. Initiating Broad-Spectrum Antibiotics

As mentioned, the decision to initiate broad-spectrum antibiotics has become commonplace in the ICU, irrespective of the severity of illness or risk for resistant pathogens. However, many patients initially suspected of having sepsis have alternative diagnoses explaining their clinical presentation. A retrospective cohort study of 2579 patients with suspected sepsis found that only 43% of patients treated with antibiotics were likely to have had an infection [10]. Given diagnostic uncertainty, the 2021 Surviving Sepsis Campaign Guidelines make the distinction that patients presenting with sepsis without shock should have a “time-limited course of rapid investigation” for potential non-infectious causes of illness, but recommend antibiotic administration be delayed no longer than 3 h [7]. These recommendations are made on the basis of studies demonstrating a reduced association of early antimicrobial therapy with mortality in patients with sepsis without shock compared to patients with septic shock [5,6,11]. One of the largest studies, a retrospective study of nearly 50,000 patient encounters, reported to the New York State Department of Health, found no significant difference in the odds of mortality with each hour delay of administration of antimicrobials in patients who did not receive vasopressors [6]. Allowing for additional time to carry out and evaluate diagnostic testing is likely to elucidate alternative diagnoses and potentially reduce unnecessary prescribing of broad-spectrum antibiotics. This practice is further supported by a single-center retrospective, observational, before-and-after study that found that a restrictive antibiotic strategy in which antibiotic therapy was only initiated after a causative organism was identified (except in patients with meningitis, severe acute respiratory distress syndrome, or septic shock) was associated with significantly lower rates of ICU-acquired extended-spectrum beta-lactamase producing Enterobacteriaceae (ESBL-E), as determined by a positive rectal swab or culture after at least 48 h from having a negative culture on admission [12]. Furthermore, the restrictive antibiotic strategy was also associated with an increased median time of being ESBL-E-free in a multivariate propensity weighted analysis.

## 3. Adverse Consequences of Broad-Spectrum Antibiotics

While there are clearly risks associated with antibiotic under-prescribing, the deleterious effects of antibiotic overuse and misuse must also be heavily considered and are drivers for additional consensus building surrounding stewardship and de-escalation efforts [13]. Ongoing use of broad-spectrum antimicrobials is a major contributor to the ever-increasing emergence of AMR [12,13,14,15]. In 2021, 4.71 million deaths were associated with AMR and 1.14 million deaths were directly attributed to AMR [16]. By 2050, it is forecasted that 8.22 million deaths will be associated with AMR and 1.91 million deaths will be directly attributed to AMR, increases of nearly 70% and 75%, respectively, per year compared to 2022. These data demonstrate the global public health threat of AMR that is outpacing the development of novel antimicrobial agents to combat it.

In addition to the global consequences of AMR, antibiotic use is also associated with several adverse effects at the patient level, including nephrotoxicity, neurotoxicity, and hematological toxicity. Drug-induced nephrotoxicity is one of the most common causes of acute kidney injury among hospitalized patients, and antibiotics are some of the most common contributors [17]. Antibiotic-induced nephrotoxicity may manifest in acute interstitial nephritis (beta-lactams, fluoroquinolones, and sulfonamides), acute tubular necrosis (vancomycin and polymixins), and Fanconi syndrome (tetracyclines). Hematologic toxicity from antibiotics often manifests as leukopenia and thrombocytopenia. Leukopenia is thought to be caused by either direct bone marrow toxicity or an immune-mediated reaction and is typically described with durations of beta-lactam therapy greater than two weeks [18]. Antibiotics may cause thrombocytopenia due to myelosuppression (linezolid) or immune mediation through various mechanisms (sulfonamides and penicillins) [19]. Neurologic toxicity may present with a variety of clinical presentations with variable severity including neuropathies (linezolid), paresthesia (polymyxins), ototoxicity (aminoglycosides), encephalopathy (beta-lactams and quinolones), neuromuscular blockade (aminoglycosides and polymyxins), and seizures (beta-lactams, quinolones, and polymyxins) [20]. These potential adverse effects, among the other consequences described in this section, should be continually evaluated when considering ongoing antibiotic use.

Broad-spectrum antibiotics also significantly impact the composition of the gut microbiome, an essential part of the immune system that protects the body from harmful pathogens and toxins [15,21]. A single-center retrospective cohort study of over 3000 mechanically ventilated, critically ill patients compared several outcomes in patients who received early antibiotics targeting anaerobes versus those who did not [22]. The most common anti-anaerobic antibiotics used were piperacillin–tazobactam (57%) followed by metronidazole (27%). Use of anti-anaerobic antibiotics was associated with reduced VAP-free days (HR 1.24, 95% CI 1.06–1.45), infection-free days (HR 1.22, 95% CI 1.09–1.38), and overall survival (1.14, 95% CI 1.02–1.28). Furthermore, when rectal swabs of 116 patients from the cohort were analyzed, the authors found that bacterial density was reduced in patients treated with anti-anerobic antibiotics (*n* = 44) compared to patients who received antibiotics that were not anti-anaerobic (*n* = 40) and patients who did not receive antibiotics prior to having a rectal swab completed (*n* = 32; *p* = 0.0016). Given that piperacillin–tazobactam was the most common anti-anaerobic agent used, the authors analyzed patients that received anti-pseudomonal coverage versus those who did not, and found no significant difference in bacterial density, VAP-free days, infection-free days, and overall survival between these two groups, therefore suggesting that anti-anaerobic activity was associated with worse outcomes. The results of this study highlight the destructive impact caused by anti-anerobic antibiotics on the host microbiome and identify an important opportunity for ADE and stewardship.

The changes to the gut microbiome from antibiotic use, along with other risk factors common to critically ill patients (previous hospitalizations, H2 blockers, proton pump inhibitor use, etc.) also contribute to the development of CDI [23]. CDI in patients in the ICU is not only associated with longer ICU and hospital lengths of stay, but significantly higher mortality and increased cost to the healthcare system [24,25]. In an effort to combat the threat of AMR, antimicrobial-associated adverse effects, deleterious changes to the gut microbiome, and the morbidity and mortality associated with secondary CDI, implementation and adherence to antimicrobial stewardship practices, specifically antimicrobial de-escalation, should be commonplace.

## 4. Antibiotic De-Escalation Definition and Rationale

ADE is defined as replacing broad-spectrum antibiotics with antibiotics with a narrower spectrum or reduced ecological impact, or stopping agents used as a part of combination therapy, including agents that may be used for dual-coverage for suspected multi-drug-resistant (MDR) organisms or used as empiric therapy for which no pathogens are isolated in culture [13]. Given the commonality of broad-spectrum antibiotic prescribing in the critically ill, ADE is a key component of successful stewardship efforts in the ICU. Antimicrobial stewardship is defined as “coordinated interventions designed to improve and measure the appropriate use of [antibiotic] agents by promoting the selection of the optimal [antibiotic] drug regimen including dosing, duration of therapy, and route of administration [26]”. 

Observational studies and small randomized controlled trials suggest that ADE and discontinuation of unnecessary antibiotics are safe practices in the critically ill, and some have demonstrated that de-escalation is associated with decreased mortality [27,28,29,30,31]. The DIANA study, an international, multicenter, prospective, observational trial, was one of the largest and most robust studies assessing the impact of ADE on clinical cure rates, among other outcomes [27]. While rates of ADE by day 3 of treatment were low (16% of 1495 included patients), there was no difference in clinical cure on day 7 or 28-day mortality between the de-escalation and no de-escalation groups. Moreover, rates of clinical cure were also higher in the de-escalation group (57.9% vs. 42.7%, *p* < 0.001); however, given the retrospective nature of this study, it is possible that de-escalation was biased toward patients with more rapid clinical improvement. More recently, a large randomized controlled trial conducted in patients with suspected sepsis in 10 United States hospitals assessed an “opt-out” protocol in which antibiotics would be discontinued after 48 to 96 h of blood culture negativity unless clinicians elected to opt out [32]. Patients in the intervention arm had 32% lower odds of continuing antibiotics, were exposed to fewer broad-spectrum antimicrobials, and did not experience additional harm, further demonstrating the safety of eliminating unnecessary antimicrobial therapy in patients with suspected sepsis.

ADE has also been linked to reductions in the development of new Gram-negative resistance and CDI. In a large multicenter retrospective cohort study of patients with sepsis who were treated with at least three days of beta-lactam antibiotics, patients who received beta-lactam de-escalation (assessed using beta-lactam spectrum score) had the lowest incidence of new Gram-negative AMR (incidence rate 1.42, 95% CI 1.16–1.68) [33]. Furthermore, a systematic review and meta-analysis of 32 studies and over 9,000,000 patient-days demonstrated that antibiotic stewardship efforts including ADE were associated with a 32% reduction in rates of CDI (IR 0.68, 95% CI 0.53–0.88) [34]. These results highlight the importance of judicious use of broad-spectrum antibiotics and initiating de-escalation efforts as soon as clinically feasible.

Even with the risks of broad-spectrum antibiotic use and benefits of ADE highlighted, the incidence of ADE in the ICU is highly variable. In the DIANA study discussed previously, only 16% of included patients had their antibiotics de-escalated by day 3 [27]. Other studies, however, report higher rates of 40–50% [35,36,37]. Taken together, these results exemplify the ongoing opportunity for improvement in the frequency of ADE, especially with the advent of rapid diagnostic testing (RDT) and other tools to aid in clinical decision making.

## 5. Timing of Assessment for Antibiotic De-Escalation

The 2021 Surviving Sepsis Campaign Guidelines suggest daily assessments for de-escalation of antimicrobials, and the 2020 European Society of Intensive Care Medicine and European Society of Clinical Microbiology and Infectious Diseases position statement on antimicrobial de-escalation in critically ill patients recommends de-escalation within 24 h of definitive culture results and antibiogram availability [7,13]. However, newer tools like RDTs may facilitate safe de-escalation before definitive culture results are available. For example, patients randomized to the multiplex bacterial polymerase chain reaction (PCR) group in the FLAGSHIP II trial received recommendations for ADE just 5 h after sample collection [38]. Several other studies have shown median times to antibiotic change based on multiplex PCR panels of 3–20 h [39,40,41]. With the availability of new tools, de-escalation should be discussed as soon as results are available while also considering both patient-specific and test-specific factors. We outline our approach to ADE in the ICU used in our clinical practice, utilizing RDTs and patient-specific risk factors, in Figure 1. This approach is based on the available literature and has evolved over the last 25 years of implementation at our institution [13,33,42,43,44,45,46].

## 6. Tools for Antibiotic De-Escalation

Recent years have seen the widespread growth of many additional tools to aid in ADE in the ICU. When available, tools such as RDTs should be used in addition to traditional Gram staining and cultures to tailor antibiotic therapy [47,48]. Of note, de-escalation tools should ideally be paired with appropriate education and antimicrobial stewardship teams to optimize their use in the clinical setting [49]. It is also imperative to consider patient-specific factors such as the severity of illness, clinical trajectory, suspected site(s) of infection, and previous culture results before performing ADE [47]. This section will describe several tools that can be used for ADE and how they can be applied in critically ill patients.

### 6.1. Rapid Diagnostic Tests

Rapid diagnostic test is an umbrella term for a large group of tests that utilize various types of technologies to deliver quick results and aid in clinical decision making. Although mainly studied to ensure that appropriate antibiotic coverage is not missed, RDTs can also play a role in ADE due to their high negative predictive value (NPV) for infection with organisms included in the test. Some of the most commonly used RDTs in clinical practice are multiplex PCR tests. Multiplex PCRs can detect prespecified pathogens and resistance genes directly from biologic samples. Tests are commercially available for many specimen types, including blood, lower respiratory tract samples, cerebrospinal fluid (CSF), and stool [47]. Negative predictive values of 92% to greater than 99% have been reported for various multiplex PCR tests from CSF, blood, and respiratory samples [50,51,52,53,54]. Implementation of multiplex PCRs in clinical practice has led to significantly faster time to appropriate antibiotics compared to traditional cultures, and in some cases higher rates of appropriate ADE [55,56]. In a multicenter, randomized controlled trial, Darie et al. protocolized de-escalation recommendations 5 h after results of the pneumonia PCR test, which led to a 45% reduction in the duration of inappropriate antibiotics without compromising time to clinical stability or mortality [38]. Due to their high NPV, one approach to the implementation of multiplex PCRs in the ICU is to de-escalate anti-MRSA and anti-pseudomonal therapy when these organisms are not detected on a tracheal aspirate PCR, for example.

Other types of RDTs include next generation sequencing, enzyme immunoassay, latex agglutination, immunochromography, MALDI-TOF, and gel electrofiltration. Table 1 describes several commercially available RDTs. Some limitations of RDTs include the inability to discern whether the organism represents an active infection or a colonizer and the limited library of organisms they detect. These are important considerations when using these tests for de-escalation purposes.

### 6.2. MRSA Nasal PCR/Cultures

Nasal swabs for MRSA screening are a minimally invasive tool frequently used for ADE purposes. The absence of nasal colonization with MRSA has been shown to predict the absence of MRSA pneumonia [57]. The 2019 American Thoracic Society (ATS) and Infectious Diseases Society of America (IDSA) Guidelines for Community-Acquired Pneumonia recommend obtaining a nasal PCR test to allow for de-escalation or confirmation of the need for continued anti-MRSA therapy [58]. While not explicitly recommended by the 2016 ATS/IDSA Guidelines for the Management of Hospital-acquired and Ventilator-associated Pneumonia, a large meta-analysis found a positive predictive value (PPV) of 35.7% and a NPV of 94.8% of MRSA nasal swabs for patients with ventilator-associated pneumonia (VAP), suggesting it is also a valuable tool for ruling out MRSA VAP [57,59]. It is important to note that in areas with a high local prevalence of MRSA, the PPV will be higher and the NPV will be lower [60]. De-escalation of MRSA-targeted therapy using MRSA nasal swabs has been shown to reduce the duration of anti-MRSA therapy in the critically ill patient population with pneumonia, with no difference in mortality or hospital length of stay [45,61]. Although the most robust evidence for de-escalation using MRSA nasal swabs exists for respiratory infections, they have been studied for use in other sites of infection. A large retrospective cohort study found that MRSA nasal screening was associated with high NPVs for wound infections (93.1%), bloodstream infections (96.5%), and intra-abdominal infections (98.6%) [62]. However, data for other sites of infection, specifically in critically ill patients, are lacking. When applying to other sites of infection, it is important to consider the pre-test probability for an MRSA infection at that site, local prevalence of MRSA, and severity of infection before de-escalating. Limitations of MRSA nasal swabs are the singular organism that they can detect, the variable PPV and NPV based on site of infection and local prevalence, and unclear effects of systemic anti-MRSA antibiotics on future MRSA swab sensitivity.

### 6.3. Biomarkers

Biomarkers are measurable substances or processes in the body that are indicative of a condition or biologic process [63]. Procalcitonin is one of the most well-studied biomarkers, particularly in sepsis due to its elevation in pro-inflammatory states, specifically in bacterial infections. The 2021 Surviving Sepsis Campaign Guidelines recommend using procalcitonin along with clinical evaluation to determine when to stop antibiotics when duration is unclear based on several studies showing decreased duration of antibiotics with no difference in safety outcomes [7,64]. In the landmark PRORATA trial including critically ill patients with suspected bacterial infection, providers were encouraged to stop antibiotics when the procalcitonin level fell to <80% of the peak value or an absolute value of <0.5 mcg/L [64]. Compared to usual care, when antibiotic initiation and duration were guided by serial procalcitonin levels patients had significantly more antibiotic-free days at day 28 (14.3 vs. 11.6, 95% CI 1.4–4.1, *p* < 0.0001). While many trials have found reduced durations of antibiotics using procalcitonin, other trials have shown no difference in antibiotic durations and even prolonged durations [65,66]. Limitations of procalcitonin include its lack of specificity for bacterial infections compared to other inflammatory processes, lack of validated cutoff points, and cost. There are several other biomarkers that have been studied in relation to acute infection. C-reactive protein (CRP) is an inflammatory marker that is often elevated during acute infections and has been studied for de-escalation purposes, but it is even less specific for bacterial infection than procalcitonin. In the recently published ADAPT-Sepsis trial, patients randomized to the procalcitonin-guided group received shorter durations of antibiotics compared to the standard care group (9.8 days vs. 10.7 days, 95% CI 0.19–1.58, *p* = 0.01); however, there was no difference in the duration of antibiotics between the CRP-guided group and standard care (10.6 days vs. 10.7 days, 95% CI −0.60–0.79, *p* = 0.79) [67]. Other biomarkers such as IL-6, HMGB1, presepsin, sTREM-1, CD64, PSP, proadrenomedullin, and pentraxin-3 are currently being studied in sepsis in hopes of continuing to improve early diagnosis and clinical interventions [68].

## 7. Antimicrobial Durations

In addition to judicious initiation of antibiotics and de-escalation, minimizing the overall duration of antibiotics is a pertinent stewardship measure. Shorter courses of antibiotics have been proven to be as effective as longer treatment durations in several randomized controlled trials for various types of infection [69,70,71,72]. Most recently, the multicenter, randomized BALANCE trial showed that 7 days of antibiotics was non-inferior to 14 days for hospitalized patients with bloodstream infections for the outcome of 90-day mortality (14.5% vs. 16.1%, 95% CI [−4.0 to 0.8]) [72]. Of note, 55% of the patients enrolled in this trial were in the ICU, and most had Gram-negative urinary tract infections or intra-abdominal/hepatobiliary infections. Observational studies have also associated shorter durations of antibiotic exposure with a decreased risk of developing resistance [42,73,74,75,76,77]. Recently, a subgroup analysis revealed a possible synergistic effect between de-escalation and antibiotic exposure duration. Compared to patients who had no change in antibiotic spectrum, de-escalated patients were observed to have a lower risk of new Gram-negative resistance development as beta-lactam exposure duration increased beyond 7 days [33]. This supports the practice of continually evaluating a patient’s antibiotic regimen for opportunities to de-escalate as treatment courses extend, particularly when an offending pathogen is not identified through rigorous culture collection. As discussed previously, biomarkers such as procalcitonin may have a role in determining the best time to discontinue antibiotics.

## 8. Potential Future Directions for ADE

### 8.1. Antimicrobial Spectrum Scoring Tools

Antimicrobial spectrum scores assign a numeric value to each antibiotic agent that corresponds with that agent′s unique spectrum of activity. These scoring systems pioneered by Madaras-Kelly et al. attempt to objectively quantify antibiotic activity using specific numeric values rather than broadly applying ordinal scale classifications [78]. Several approaches to spectrum scoring have been described, all of which delineate de-escalation as a reduction in score during early treatment (day 2 versus day 4) or cumulatively over a patient’s total antibiotic exposure during hospitalization [43,79,80,81]. The cumulative approach could limit the risk of misclassifying ADE in patients who receive longer courses or multiple antibiotic regimens during their hospitalization. Studies utilizing spectrum scores have associated ADE with fewer antibiotic days, decreased length of stay, decreased episodes of CDI, and a reduction in new Gram-negative resistance [33,43,82]. In addition, higher global spectrum scores, indicating continued broad-spectrum antibiotic exposure, were associated with increased hospital mortality in a multicenter observational study [83]. While spectrum scores have begun to establish their role in analyzing ADE and antibiotic utilization data, they offer a variety of potential applications that can be leveraged in the future. For example, spectrum scores could be integrated into electronic health records to identify patients who may be on unnecessarily broad antibiotics for a particular type of infection to facilitate ADE [84].

### 8.2. Artificial Intelligence

As artificial intelligence (AI) technologies advance, offering patient-tailored interventions to implement de-escalation effectively and efficiently will be routinely available. Machine learning methods falling under the umbrella of either neural networks or gradient boosting have been utilized to predict antibiotic susceptibility and resistance, develop personalized antibiograms, forecast the effectiveness of antibiotic treatment, and optimize initial dosing of antibiotics [85]. Multiple machine learning approaches are employed to develop, train, and test model performance during the process of identifying an architecture that provides the greatest predictive accuracy. Once the final machine learning prediction model has been established, implementation as a decision-support tool and evaluation of clinical utility is critical. Machine learning models to predict personalized antibiograms that estimate the probability that common antibiotic regimens will provide activity against infections at the point in time empiric antibiotics are chosen can significantly narrow broad-spectrum prescriptions and facilitate increased de-escalation practices compared to clinician selections [86,87]. Furthermore, implementation of AI modeling aiming to modify empiric therapy with greater precision has been associated with reduced hospital length of stay and risk of complications such as CDI [88]. The ability of AI-based models to analyze clinical data and integrate into the electronic health record will aid the refinement of antibiotic treatment regimens and enhance stewardship efforts.

## 9. Conclusions

As AMR continues to increase over time, optimizing antimicrobial stewardship and de-escalation practices is increasingly prudent, especially in vulnerable patient populations like the critically ill. As it stands, antimicrobial stewardship is not performed optimally in all critically ill patients, despite data continuing to support its safety. However, with the advent of RDTs, de-escalation can occur sooner than ever before if these tools are leveraged to their full potential. All ICU clinicians should continually assess the appropriateness of antimicrobial therapy and seek opportunities to de-escalate therapy or discontinue unnecessary antimicrobials. We outline several key takeaway points from this review in Table 2 to consider when incorporating ADE into clinical practice.

## Figures and Tables

**Figure 1 antibiotics-14-00467-f001:**
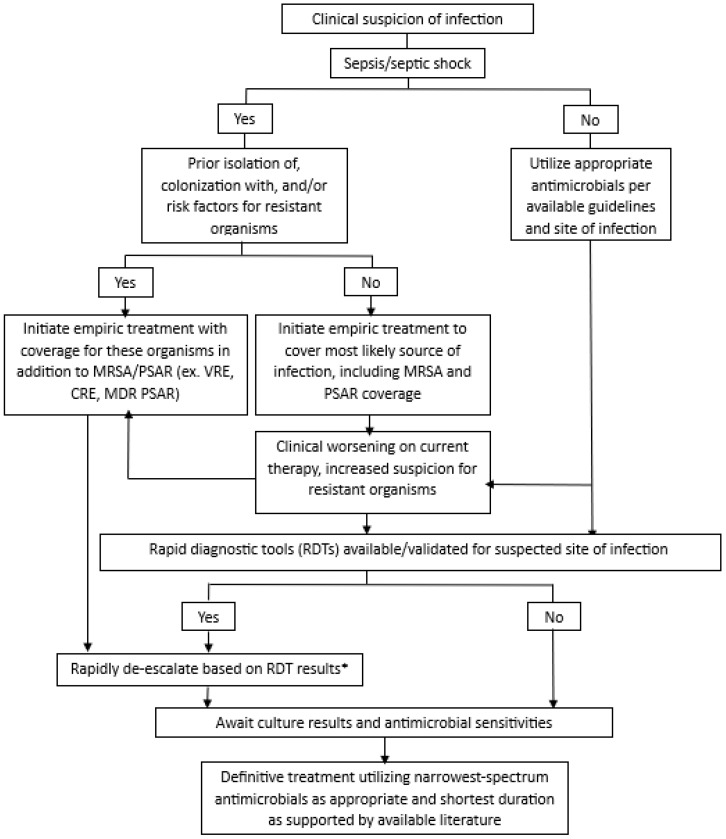
Approach to antimicrobial de-escalation in critically ill patients. * Before de-escalating based on the RDT, must also consider severity of illness, clinical trajectory, suspected site(s) of infection, previously isolated organisms, and bacterial targets of the RDT; ^ⱡ^ if clinically indicated based on patient factors and local pathogen prevalence; MRSA: methicillin-resistant *Staphylococcus aureus*; PSAR: *Pseudomonas aeruginosa*; VRE: vancomycin-resistant *Enterococcus*; CRE: carbapenem-resistant *Enterobacterales*; MDR-PSAR: multi-drug-resistant *Pseudomonas aeruginosa*.

**Table 1 antibiotics-14-00467-t001:** Select commercially available rapid diagnostic tests.

Type of Test	Subtype	Commercially Available Test Examples	Specimen	Number of Bacterial Targets *	Resistance Detection
Nucleic acid amplification test	Multiplex PCR	BioFire FilmArray Pneumonia Panel	Respiratory	18	√
		Unyvero LRT	Respiratory	20	√
		Biofire FilmArray BCID	Blood	26	√
		Unyvero BCU	Blood	26	√
		BioFire FilmArray Gastrointestinal Panel	Stool	13	
		BioFire FilmArray Meningitis Encephalitis Panel	CSF	6	
Next-generation sequencing	Metagenomic NGS	Karius	Blood	770	√
Nanoparticle probe technology	-	Verigene BC-GN	Blood	8	√
Enzyme immunoassay	Enzyme-linked immunosorbent assay	Techlab C. Diff Quik Chek Complete	Stool	1	
Mass spectrometry	MALDI-TOF	Vitek MS	Blood	1095	
Fluorescent in situ hybridization	-	Accelerate Pheno	Blood	14	√

* At the time of submission for publication. Target/pathogen lists for certain tests are continuously updated. PCR: polymerase chain reaction; CSF: cerebrospinal fluid; NGS: next-generation sequencing; MALDI-TOF: matrix-assisted laser desorption ionization time-of-flight.

**Table 2 antibiotics-14-00467-t002:** Key takeaway points.

Antibiotic De-Escalation: Key Takeaways
Deaths associated with and attributed to antimicrobial resistance are expected to increase over the coming years.
ADE has been demonstrated to be safe in several observational trials and small randomized controlled trials of critically ill adults.
ICU practitioners should continually assess the appropriateness/need for antimicrobial therapy. ADE should take place within 24 h of microbiologic culture results and antibiogram receipt, if not sooner based on rapid diagnostic tests.
Rapid diagnostic tests, MRSA nasal PCR, and biomarkers may provide additional data to prompt more efficient ADE.
Shorter courses of antibiotics have been proven to be as effective as longer durations in several randomized controlled trials. Minimizing antibiotic durations may be leveraged as a key stewardship effort.
As technology continues to advance, including artificial intelligence, it should continue to be leveraged to support ADE and stewardship efforts.

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
