# Peer review of "Antibiotic De-Escalation in the Intensive Care Unit: Rationale and Potential Strategies"

_antibiotics, 2025, doi:10.3390/antibiotics14050467_

Round 1
Reviewer 1 Report
Comments and Suggestions for Authors
Dear Authors,
I have read your paper with care and interest. It is an interesting and very well written narrative review on antibiotic de-escalation in the ICU. The article will be an interesting read for both clinicians and students. It also provides future perspectives on the topic that might stimulate future research in the field.
I have the following suggestions to improve clarity and readability:
- in the "Introduction" section, p. 2, lines 46-48, I suggest rewording the statement presenting the contents of the paper by adding an overview of the topics covered, such as tools for ADE, future directions, etc. This will help the reader to get an overview of contents before reading further;
- p. 2, lines 81-83, deaths associated with AMR are reported; however, the data is from 2019 (6 years old) and only derived from the systematic review of Murray et al. [16]. I would suggest integrating these data with more recent statistics, also from global/regional health authorities;
- in the "Introduction" section, p. 1-2, lines 41-44, it is stated that "there are many possible adverse consequences of empiric broad-spectrum antibiotics, including Clostridioides difficile infection (CDI), acute kidney injury, haematological or neurological toxicity, disruption of the gut microbiome, and the development of antimicrobial resistance (AMR)", but in the section "Adverse Consequences of Broad-Spectrum Antibiotics" you discuss AMR, the gut microbiome, and CDI infection. I suggest you consider discussing the other topics you mentioned as well to give the reader a comprehensive overview;
- the section "Definition and Rationale for Antibiotic De-Escalation" begins with a definition of antimicrobial stewardship. As an optional suggestion, I would suggest considering introducing ADE first and then mention that it is part of antimicrobial stewardship practices;
- at the end of the section "Definition and Rationale for Antibiotic De-Escalation", p. 4, lines 154-156, the statement "these results exemplify the ongoing opportunity for improvement" could be further clarified;
- I found Figure 1 very interesting, clearly designed and useful for clinical practice; however, on page 4, line 169, you introduce it by stating "we outline our approach". I suggest you add more detail on how Figure 1 was derived - is it currently used in your clinical practice? If so, are there any data on its effectiveness? Was it developed following a literature review? In this case, could you provide some supporting references for items? Including these details will give the reader the supporting background to the figure.
- I suggest you add a supporting reference for the statements on page 5, lines 179-185.
- in the "Rapid Diagnostic Tests" section, you highlight the high negative predictive value (NPV) of RDTs. I suggest you add some examples of NPV values and supporting references, as you did with MRSA nasal swabs;
- I really appreciated the section on "Potential Future Directions for ADE". I encourage the authors to deepen the perspectives presented, especially on AI. A considerable number of papers have been published in the literature on the topic of AI-driven antibiotic therapy optimisation, and the paper would benefit from some more details on this important emerging topic;
- in the Abstract, the first lines up to almost halfway through, emphasise the importance of conducting ADE, while little attention is given to the rest. As this part is already thoroughly described in the text, I would suggest rewriting the abstract to focus on a concise summary of all the topics covered;
- as an additional suggestion, I would suggest to consider including a table or a paragraph with the key take-home messages of your review (e.g., "All ICU clinicians should continually assess the appropriateness of antimicrobial therapy", etc.).
I hope that my suggestions will be helpful in improving your work.
Author Response
Please see attached file for responses to reviewer comments.

Reviewer 2 Report
Comments and Suggestions for Authors
This review explained the importance of antibiotic de-escalation in the intensive care unit. Review highlighted the drawbacks of using broad-spectrum antibiotics and emphasized the necessity of rapid diagnostic tests in clinical decision making. At the end also suggested the use of antimicrobial spectrum tools and artificial intelligence towards effective antibiotic stewardship. Review is well structured and I don't have any concerns. I have a suggestion for authors to discuss possible limitations of antibiotic de-escalation if any.
Author Response
This review explained the importance of antibiotic de-escalation in the intensive care unit. Review highlighted the drawbacks of using broad-spectrum antibiotics and emphasized the necessity of rapid diagnostic tests in clinical decision making. At the end also suggested the use of antimicrobial spectrum tools and artificial intelligence towards effective antibiotic stewardship. Review is well structured and I don't have any concerns. I have a suggestion for authors to discuss possible limitations of antibiotic de-escalation if any. |
We appreciate your positive feedback on the review. Throughout the article, we discuss the potential drawbacks to antibiotic under-prescribing in patients with sepsis and septic shock (ultimately increases mortality). In our opinion, this is one of the largest limitations to antibiotic de-escalation. We also discuss that antimicrobial de-escalation is not always done and that there are no standardized practice guidelines for practitioners to follow. While we did not add a specific limitations section to the article, we believe many of the potential limitations are discussed throughout. Please let us know if you see these items to be sufficient. Again, we appreciate your review. |
Round 2
Reviewer 1 Report
Comments and Suggestions for Authors
Dear Authors,
Thank you for your revisions. All my suggestions have been addressed.
Congratulations on your work, which I believe will be an interesting read for both clinicians and students. I have no further suggestions.